# Use of Cold-Pressed Sunflower Cake in the Concentrate as a Low-Input Local Strategy to Modify the Milk Fatty Acid Profile of Dairy Cows

**DOI:** 10.3390/ani9100803

**Published:** 2019-10-14

**Authors:** Idoia Goiri, Izaro Zubiria, Hanen Benhissi, Raquel Atxaerandio, Roberto Ruiz, Nerea Mandaluniz, Aser Garcia-Rodriguez

**Affiliations:** NEIKER-Granja Modelo de Arkaute, Apdo. 46, 01080 Vitoria-Gasteiz, Spain; izaro23@hotmail.com (I.Z.); hanening@gmail.com (H.B.); ratxaerandio@neiker.eus (R.A.); rruiz@neiker.eus (R.R.); nmandaluniz@neiker.eus (N.M.)

**Keywords:** oilseed, alternative feedstuffs, conjugated linoleic acid, cattle, fat

## Abstract

**Simple Summary:**

Consumers demand healthier dairy products. Supplementing plant lipids, rich in poliunsaturated fatty acids, results in improved milk fatty acid profile, but these oils could enter into competition with human food needs and compromise animal performance. The aim of this study was to test the feasibility of formulating cold-pressed sunflower cake (CPSC, high-fat by-product) in a dairy cows’ concentrate to improve milk fatty acid profile. Cold-pressed sunflower cake increased total trans-mono unsaturated fatty acids (21%), total conjugated linoleic acid (31%), and polyunsaturated fatty acids to saturated fatty acids ratio (18%), but did not affect milk production, digestibility, intake, and milk composition. However, reduced fat yield (9%) and fat corrected milk (7%) were observed. Feeding CPSC improved overall acceptability of milk by improving flavor. In conclusion, CPSC could modify milk FA profile without observing a detrimental effect on digestibility, production performance, or milk acceptance. Adopting feeding systems based on the use of cheaper and local alternative feedstuffs rich in polyunsaturated fatty acids would represent a good strategy to change milk fatty acid profile and contribute the promotion of low-input production systems.

**Abstract:**

Cold-pressed sunflower cake (CPSC) is a cheap by-product of oil-manufacturing. Supplementing diets with CPSC, rich in fat and linoleic acid, could be an effective tool for increasing healthy fatty acids (FA) in milk. To test this hypothesis, 10 cows were used in a crossover design with two experimental diets fed during two 63-day periods. Cows’ milk production was recorded and samples were taken for fat, protein, lactose, and for FA composition analysis. Dry matter intake (DMI) and dry matter apparent digestibility (DMD) were estimated using two markers. Milk acceptance test was carried out. CPSC decreased milk C12:0 (10%, *p* = 0.023) and C16:0 (5%, *p* = 0.035) and increased C18:1 cis-12 (37%, *p* = 0.006), C18:1 trans-11 (32%, *p* = 0.005), C18:2 cis-9 cis-12 (13%, *p* = 0.004), and cis-9 trans-11 CLA (35%, *p* = 0.004). CPSC increased total trans-monounsaturated FA (21%, *p* = 0.003), total CLA (31%, *p* = 0.007), and PUFA:SFA ratio (18%, *p* = 0.006). CPSC did not affect milk production, DMD, DMI and milk composition, but reduced fat yield (9%, *p* = 0.013) and FCM (7%, *p* = 0.013). CPSC improved milk overall acceptability. In conclusion, CPSC could modify milk FA profile without a detrimental effect on digestibility, production performance, or milk acceptance.

## 1. Introduction

In the last decade, a significant research effort has been focused towards modifying ruminants’ milk fat composition in order to increase the concentration of fatty acids (FA) with positive effects on human health [1]. Feeding lipids has received a particular attention as a mechanism to manipulate FA profile of ruminant derived products [1]. In this sense, supplementing diets with plant lipids containing high proportion of unsaturated fatty acids (UFA) has proven to be an effective strategy for increasing healthy FA in milk [1]. Some studies, however, demonstrated that this nutritional strategy could affected negatively diet digestion and rumen fermentation due to a negative effect of dietary UFA on the growth and activity of rumen microorganisms [2]. Further research also indicated that these disturbances largely depended on the amount, type, and physical form of lipid supplement [3].

Adopting feeding systems based on the use of cheaper and local alternative feedstuffs rich in UFA would represent a good strategy to change FA profile of milk and dairy products and contribute to the promotion of low-input production systems. Cold-pressed sunflower cake (CPSC) is a cheap by-product of biodiesel production which can be obtained on-farm after simple mechanical extraction of the oil. They are obtained by simple passing of the seed through a continuous screw press that generates only mechanical heat. In addition to its high content of protein, CPSC has higher crude fat content than those of conventional solvent and expeller meals (up to 230 g/kg compared to 30 and 100 g/kg, respectively; [4]), which make it an attractive energetic feedstuff for lactating cows. Moreover, CPSC contains a high proportion of linoleic acid [5], which makes it a promising lipid supplement to change milk FA profile.

Numerous experiments have been carried out to assess the effects of sunflower oil on rumen fermentation process and milk FA profile [6], but few data are available concerning the impact of diet supplementation with CPSC. In this sense, a recent study has shown that feeding CPSC instead of palm fat to ruminants might improve the rumen FA profile, mainly by reducing medium-chain saturated FA (SFA) and promoting vaccenic acid (VA) and rumenic acid (RA) accumulation in vitro [7]. However, changes on rumen FA profile might be associated with impaired rumen function, depending on the CPSC inclusion level in the diet [4]. Feeding CPSC has also proven very effective in reducing saturated FA and increasing VA and RA in milk of dairy sheep [5]. Finally, although a more unsaturated milk FA profile could have a beneficial effect in consumer’s health, it sometimes has been related to altered milk sensory characteristics [8].

Therefore, the objective of the present study was to test the feasibility of using CPSC in the formulation of a concentrate for dairy cows, totally replacing hydrogenated palm fat, for improving milk FA profile. In addition, we must ensure that this feeding strategy does not compromise productive performance and related traits or milk sensory characteristics.

## 2. Materials and Methods

All experimental procedures were performed in accordance with the European Union Directive (2010/63/EU) and Spanish Royal Decree (RD 53/2013) for the protection of animals used for experimental and other scientific purposes, and approved by the ethics committee (NEIKER-OEBA-2015-011).

### 2.1. Animals and Experimental Design

All cows were kept at the experimental research farm of Fraisoro Farm School (Zizurkil, Spain) in loose housing conditions. A total of 4 lactating Holstein and 6 Brown Swiss dairy cows were paired based on breed, parity, days in milk (DIM), and milk yield during a 2-week covariate period. Average DIM, body weight (BW), and milk yield of the cows before initiation of the experiment were (mean ± SD): 106 ± 37 d, 600.6 ± 63.8 kg, and 26.3 ± 6.3 kg, respectively. All cows were fed the control diet (Table 1) during the covariate period and randomly assigned (within pair) to the control (CTR) or experimental diets (CPSC) in a cross over design. Diets were formulated based on INRA [9] recommendations taking into account milk production and days in milk.

The treatment was formulated into concentrate. The percentage of fat of the two concentrates was the same (5.6% of the concentrate; Table). The CTR concentrate had hydrogenated palm fat and the CPSC had sunflower oil contained in it as fat source. We formulated the concentrate taking into account the forage to concentrate ratio normally observed in the experimental farm of Fraisoro, and taking into account not to include a fat content in total diet more than 5% that can be related with depressed milk fat content or milk production, especially with fat sources rich in polyunsaturated fatty acids.

Each period lasted for 63 d; the first 14 d were allowed for adaptation and measurements were taken during the following 49 d. The data frame of the experimental design can be seen in Figure 1.

The concentrates were formulated to provide similar amounts of energy, CP, and fat (Table 1). The FA profile of the concentrates can be seen in Table 2 (Table 2). Cows within a pair received the same amount of concentrate in individual troughs offered at three different times per day, but they had free access to a basal roughage mixed ration. For an interval of 30 d between each experimental period, cows were fed on mixed roughage ad libitum and CTR concentrate to avoid a carryover effect.

Cows were milked with an automated milking system (AMS, DeLaval, Tumba, Sweden, 2004). All cows had free access to the AMS 22.5 h/d (a total of 1.5 h was dedicated to the automatic cleaning of the system). Cows were granted milking permission after 11 h from previous milking, unless a milking failure occurred, in which case cows would be granted permission to be milked again immediately. In general, for any particular cow, when the time elapsed since last milking was more than 12 h during the day, that cow would be fetched and forced to visit the AMS.

### 2.2. Sampling and Measurements

#### 2.2.1. Milk Production and FA Profile

Individual daily milk production was recorded at each milking by the AMS. Individual milk samples were collected from the AMS at each milking on the last day of the covariate period and during weeks 5, 7, and 8 of each experimental period, and stored with azidiol (3.3 mL/L) at 4 °C for fat, protein, and lactose analysis. In week 9 of the experimental period, additional individual milk samples were collected from the AMS at each milking for FA profile determinations. Additionally, a composite milk sample (9 kg) from each treatment was collected into stainless steel milk cans for organoleptic characteristics determination at the end of period 2 of the cross-over design.

#### 2.2.2. Intake and Apparent Digestibility of Nutrients

Basal roughage and concentrates were sampled weekly. Beginning in week 5 and over 10 days, cows received twice daily, at 06:00 h and 18:00 h, 20 g of Cr_2_O_3_ mixed with the concentrate. After the 7-day standardization period, the fecal sample collection was started, lasting 3 consecutive days. Fecal grab samples (approximately 400 g/sample) were collected from each cow from the rectum at 12-h intervals. 

#### 2.2.3. Feeding Behavior

In week 7 individual feeding behavior data was recorded during two consecutive days. One observer observed the flock at a reasonable distance, and every effort was made not to disturb the flock. Eating, ruminating, and other activities of the 10 animals were recorded at 5-min intervals.

Animals were weighed on the last day of the covariate period and on the last day of the experimental period.

### 2.3. Sample Handling and Laboratory Procedures

#### 2.3.1. Feed and Feces

Roughage and concentrate were dried in a forced-air oven and fecal grab samples were freeze-dried (Christ Alpha 1-4 LD Plus, Fisher Bioblock Scientific, Madrid, Spain), ground through a 1-mm sieve, and composited by period and animal. Dry matter (method 934.01) and N (method 984.13) contents were determined following AOAC [10]. Neutral detergent fiber was determined by the method of Van Soest [11] with use of an alpha amylase, but without sodium sulphite in the neutral detergent solution, and was expressed free of ash. Acid detergent fiber, expressed exclusive of residual ash, was determined by the method of Robertson and Van Soest [12]. Fat content was determined without hydrolysis by the automated Soxhlet method (Soxtec System HT 1043 Extraction Unit, Madrid, Spain) using hexane for 6 h as solvent. Starch content was measured by polarimetry [13]. Acid-insoluble ash (AIA) contents of feeds and feces were determined gravimetrically after drying, ashing, boiling of ash in hydrochloric acid, filtering and washing of the hot hydrolysate, and re-ashing [14]. Feces were analyzed for Cr by atomic absorption spectrometry.

Fatty acid methyl esters (FAME) of lipid in both concentrates were prepared in a one-step extraction-trans-esterification procedure using chloroform and 20 mL/L sulphuric acid in methanol [15]. Methyl esters were separated and quantified with a gas chromatograph (Agilent 7890A GC System, Santa Clara, CA, USA) equipped with a flame-ionization detector and a 100-m fused silica capillary column (0.25 mm i.d., 0.2-μm film thickness; CP-SIL 88, CP7489, Varian Ibérica S.A., Madrid, Spain) and hydrogen as the carrier gas (207 kPa, 2.1 mL/min). Total FAME profile in a 2 μL sample volume at a split ratio of 1:50 was determined using the temperature gradient program described in [15]. Peaks were identified based on retention time comparisons with commercially available standard FAME mixtures (Nu-Chek Prep., Elysian, MN, USA; and Supelco37component FAME mix, Sigma-Aldrich, Madrid, Spain).

#### 2.3.2. Milk

Milk fat, protein, and lactose contents were analyzed by near-infrared spectroscopy (Foss System 4000, Foss Electric, Hillerød, Denmark; ILL, Lekunberri, Spain). Milk samples for FA determinations were composited by animal relatively to milk production and were stored and preserved at −20 °C with azidiol. For analysis of milk FA profile, milk fat extraction was carried out according to ISO 14156 [16], methylated according to ISO 15884 [17], and analyzed using gas chromatography. The upper phase was injected into a gas chromatograph (Varian 3800) equipped with a capillary column (Cp-sil 88 to over 50 m) and the FID detector. Working conditions were set according to standard [18]. Carrier gas, nitrogen with a pressure of 14 psi, was used and the injector temperature was 250 °C. In order to isolate the exact and complete recovery of FA (particularly short-chain type) temperature program proposed by Kramer et al. [19] was used: temperature of 45 °C (4 min), increase in temperature of 13 °C per minute up to a temperature of 175 °C (27 min), increase in temperature of 4 °C per minute up to 215 °C (35 min).

### 2.4. Milk Sensory Acceptance Test

Raw milk was pasteurized at 72 °C for 30 s using a continuous plate heat exchanger (ATA Tecnología Alimentaria, Irun, Spain). A triangle test was used to determine the consumers’ ability to distinguish differences between samples. Eighty-one panelists evaluated 4 milk samples per treatment in private booths. Panelists were served 2 sets of samples in which the reference was either milk from the control diet or cold-pressed sunflower cows. In every set, one sample was identical to the reference and one was different. After being presented with a sample set, panelists were asked to identify the sample that tasted the same as the reference. The acceptance test was carried out using a non-trained sensory panel of women and men, regular consumers of cow milk (*n* = 58). A 9-point line scale was used, with ‘1’ being the lowest and ‘9’ representing the highest intensity, for the attributes of appearance, flavor, odor, texture, and overall acceptability.

### 2.5. Calculations and Statistical Analysis

Milk fat, protein, and lactose concentrations were calculated as weighted average of milking data.

Concentrate intake was measured as the difference between the quantities offered and refused. Fecal output, roughage DM intake (DMI), and apparent DM digestibility (DMD) were calculated using Cr_2_O_3_ as external marker and AIA as internal marker following the formula proposed by Cochran and Galyean [20].

A meal was considered to be a sequence of at least two successive eating observations. Behavior activities, DMI and DMD estimates, and SCFA concentrations were averaged by cow and period.

3.5% FCM was calculated as: 0.4318M + 16.23F, where M is milk production (kg) and F is milk fat (kg).

Each cow was considered as the experimental unit. Milk yield, FCM, milk fat and protein contents, and milk fat and protein yield (*n* = 10) were analyzed using the MIXED [21] procedure for repeated measures [22] and assuming a covariance structure fitted on the basis of Schwarz’s Bayesian information model fit criterion. The statistical model included fixed effects of concentrate, breed, period, sequence and week, and the initial record measured at week 0 (covariate). The model included the random effect of cow within pair. Feeding behavior data, DMI, DMD, milk FA profile (*n* = 10), were analyzed using the previous statistical model but without considering the effect of week or including a covariate. Sensorial data (*n* = 58) were analyzed with the previous statistical model but without considering the effect of period. Least squares means for treatments are reported. For the main parameters, treatment means were separated using a Tukey test and for milk FA profile Bonferroni adjustment was used. Significant effects were declared at *p* < 0.05.

## 3. Results

### 3.1. Intake, Apparent Digestibility of Nutrients, and Feeding Behaviour

The effects of dietary treatments on intake, apparent digestibility of nutrients and feeding behavior are shown in Table 3. The total DMI (*p* = 0.813), roughage DMI (*p* = 0.867), the daily intakes of N (*p* = 0.366), starch (*p* = 0.364), NDF (*p* = 0.947), or ADF (*p* = 0.921) were not affected by the inclusion of CPSC in the diet. Nor were there any changes in apparent digestibility of DM (*p* = 0.381), starch (*p* = 0.591), NDF (*p* = 0.503), or ADF (*p* = 0.342) although a tendency was found for a greater N apparent digestibility when CPSC was fed (*p* = 0.086).

Feeding CPSC did not affect eating time (*p* = 0.233), rumination time (*p* = 0.177), chewing time (*p* = 0.313), or idling time (*p* = 0.129). Non-significant differences were found between treatments in terms of rumination time per DMI (*p* = 0.579) or chewing time per DMI (*p* = 0.137). Similarly, feeding CPSC did not affect number of meals per day (*p* = 0.69) or meal duration (*p* = 0.129).

### 3.2. Milk Yield, Milk Quality, and Feed Efficiency

The effect of feeding CPSC on milk yield and composition, and feed efficiency can be seen in Table 4. Feeding CPSC did not affect yields of milk (*p* = 0.233), crude protein (*p* = 0.267), or lactose (*p* = 0.537), but reduced yields of fat by 9% (*p* = 0.013) and FCM by 7% (*p* = 0.013). Milk composition measured as crude fat (*p* = 0.101), crude protein (*p* = 0.267), or lactose (*p* = 0.342) was not affected by feeding CPSC. In terms of feed efficiency, feeding CPSC did not affect milk/DMI (*p* = 0.472), N yield as % of N intake (*p* = 0.694) but tended to decrease 3.5% FCM/DMI by 7% (*p* = 0.092).

### 3.3. Milk Fatty Acid Composition

Table 5 shows the SFA profile of milk in cows fed either the CTR or the CPSC diets. The percentages of most short and medium chain SFA were not affected by the dietary treatments, except for C12:0 which was a 10% lower in milk fat of CPSC fed-cows (*p* = 0.023). Feeding CPSC induced a significant decrease of 5% in C16:0 (*p* = 0.035), and a concomitant increase of 6% in C18:0 (*p* = 0.025). Total SFA did not differ (*p* = 0.238) between CTR and CPSC-fed cows.

Responses of milk fat mono unsaturated (MUFA) and polyunsaturated (PUFA) to dietary treatments are reported in Table 5. The CPSC diet did not affect milk content of C18:1 cis-9 (*p* = 0.983) or C18:1 cis-11 (*p* = 0.953), but increased C18:1 cis-12 by 37% (FA, *p* = 0.006). Total cis MUFA remained statistically unmodified when CPSC was fed (*p* = 0.877).

Feeding CPSC increased C18:1 trans-11 content by 32% (*p* = 0.005), without affecting C18:1 trans-10 level (FA, *p* = 0.215) or C18:1 trans-10/trans-11 ratio (*p* = 0.173). Total content of trans-MUFA was 21% higher in milk fat of CPSC-fed cows (*p* = 0.003). Total MUFA, however, did not differ between treatments (*p* = 0.570). 

As shown in Table 5, the use of CPSC as an alternative feed increased milk fat C18:2 cis-9 cis-12 by 13% (*p* = 0.004), cis-9 trans-11 CLA by 35% (*p* = 0.004). Total content of CLA was a 31% higher in milk fat of cows receiving CPSC diet (*p* = 0.007). The percentages of long-chain n-3 PUFA (*p* = 0.558) were not affected by the dietary treatments, but 24:1n9 increased 33% with the use of CPSC (*p* = 0.019). The ratio of n6:n3 in milk did not differ between treatments (*p* = 0.279), whereas the ratio of PUFA:SFA was 18% greater in CPSC (*p* = 0.006).

### 3.4. Pasteurized Milk Perceptibility and Sensory Properties

Results from the triangle test shown that feeding CPSC compared to CTR resulted in a milk perceptibly different (*p* < 0.001). Effect of feeding CPSC on pasteurized milk sensory properties can be seen on Table 6. Feeding CPSC improved overall acceptability by 0.6 point out of 9 (*p* = 0.003) by improving flavor 0.6 point out of 9 (*p* = 0.008). Non-significant differences were found for appearance (*p* = 1), odor (*p* = 0.702), or texture (*p* = 0.629).

## 4. Discussion

The increased need for energy, the fluctuations in world fuel prices and the growing concerns about emissions and climate change, have prompted governments to encourage the development and utilization of locally available, environmentally-friendly, and renewable sources of energy, such as biodiesel. In large-scale biodiesel production seeds are treated chemically in order to extract as much oil as possible. Opposite small-scale production on farm is based in a physical cold-pressure of the seeds, in the case of this study sunflower seeds, obtaining oil and a vegetable cake (CPSC) as a by-product. The use of the obtained oil as biodiesel in the farm machinery and the cake as feedstuff contributed to sustainably reduce the inputs of the farms. The obtained cake, compared to conventional cakes obtained in large scale biodiesel production, has higher residual oil content and a very interesting FA profile making it a promising energetic feedstuff for livestock. In this context, the hypothesis of this study was that local alternative feedstuffs rich in UFA, like CPSC, could be used in the formulation of a concentrate for dairy cows, totally replacing hydrogenated palm fat, with the goal of favorably modifying milk FA profile. Nevertheless, it is important to address that this feeding strategy must not have a negative effect on animal performance or on dairy products quality.

### 4.1. Milk FA Profile

Feeding CPSC induced a decrease in the percentages of C12:0 and C16:0 and an increase of C18:0 in milk fat. This is in agreement with results of recent studies in dairy cows fed sunflower cake [23] and dairy sheep supplemented with CPSC [5] or sunflower oil [6]. The lower content of C12:0 and C16:0 in milk can be primarily accounted for their lower contents in CPSC diet, but also for the increased supply of long-chain FA, mainly linoleic acid, that might have a potential inhibitory effect on de novo synthesis of short and medium chain SFA in the mammary gland of CPSC-fed cows [24]. The higher content of C18:0 can be attributed to the increased supply of linoleic acid for C18:0 synthesis in the rumen, and the greater ruminal escape of C18:0 in the same CPSC-fed dairy cows [25].

High intakes of C12:0 and C16:0 could be related to an increased risk of cardiovascular disease and development of the metabolic syndrome [26]. Moreover, although SFA are generally considered unhealthy due to their hypercholesterolemic effects, C18:0 has been shown to have a beneficial impact on blood lipids [27]. Thus, the effect of feeding CPSC on the levels of these SFA would have a positive effect on the nutritional value of milk fat.

The percentage of C18:1 cis-9 in milk remained unmodified with CPSC, which is consistent with the small-to-negligible changes observed in the proportion of this FA in the milk of cows fed sunflower seed or oil [28]. The C18:1 trans-11 (VA) increased in response to CPSC feeding. This response agrees with the increased VA concentration observed in rumen of the same cows fed CPSC [25] and is in line with the findings of Benhissi et al. [7] who reported that, due to its elevated linoleic acid content, CPSC may inhibit the last step of biohydrogenation, increasing the ruminal outflow of VA and enhancing the deposition of this healthy FA in ruminant-derived products. Surprisingly, CPSC-induced variation in milk fat VA was not accompanied by relevant differences in C18:1 trans-10, a FA with an uncertain involvement in human coronary heart disease [29]. The level of C18:1 trans-10 tends to increase markedly with certain nutritional strategies based on diet enrichment with linoleic acid rich lipids, and it can even exceed that of VA, particularly in dairy cows [26]. Nevertheless, this shift in C18:1 trans-10/C18:1 trans-11 ratio—a clear indicator of altered ruminal environment and biohydrogenation pathways—was not observed in the current study. Reasons for this discrepancy between C18:1 trans-10/C18:1 trans-11 ratio responses to supplementary lipids remain uncertain, but it could be attributed to the form of lipid supplements [6]. It is possible that feeding lipids, such as CPSC, may produce lower alterations in the rumen environmental conditions and biohydrogenation pathways than free fats or oils.

In relation to C18:2 isomers, the increase in C18:2 cis-9 cis-12 in milk fat of CPSC fed-cows evidenced that not only C18:1 trans saturation, but also C18:2 cis-9 cis-12 hydrogenation was constrained by increased intake of PUFA [7]. The milk content of C18:2 cis-9 trans-11 CLA rose in response to CPSC feeding, which agrees with the greater increase in the accumulation of its precursor (VA) for mammary Δ9-desaturation [30]. A recent study in dairy sheep fed with diet enriched with high levels of CPSC supplies similar results [5]. These data also agree with others reported by Collomb et al. [31] and Rego et al. [32] in milk fat from cows supplemented with sunflower seed or oil.

Overall, feeding CPSC failed to affect the n6:n3 ratio, possibly due to its low content of linolenic acid, but modified the fatty acid profile of milk toward a higher (more beneficial) PUFA:SFA ratio and increased to a certain extent the content of some interesting PUFA with positive effects on human health, a change that might improve the nutritional quality of milk fat.

### 4.2. Productive Performance and Feeding Behavior

The lack of detrimental effects on animal production parameters or milk yield due to CPSC supplementation agrees with the maintained DMI and DMD observed and the lack of effects on feeding behavior, and is consistent with other studies in dairy ewes [5]. The lack of effects on feeding behavior is important and helps us understanding the lack of effects on intake and digestibility, since previous studies have been observed that intake and digestibility were highly related to feeding behavior measurements, such as rumination time [33]. In dairy cow studies, it has been described that supplementation of free oils rich in linoleic acid, such as sunflower oil, can reduce milk fat yield by reducing secretion of milk FA with fewer than 18 carbons even at a level less than 3% of total fat [34]. In the present study, however, milk production and milk fat proportion were not decreased and only a slight decrease in milk fat yield and FCM was observed in cows fed CPSC, with an inclusion of 4.5% of total fat. This could be related with the physical form of the lipid supplement [35]. It is possible that lipids in the cake are not as accessible as free oil lipids to interact with ruminal microorganisms and thus may produce lower alterations in the rumen environmental conditions than free fats or oils. Nevertheless, even a small decrease in milk fat yield and FCM is not desirable. Thus, these results could suggest that the cake inclusion level in the ration is in the upper limit for dairy cows under the feeding regime studied in this trial.

### 4.3. Curd Sensory Acceptance Test

The quality and composition of milk are influenced by several dietary factors like dietary FA profile. In this sense, when evaluating the suitability for formulation of alternative feedstuffs rich in lipids, and specifically rich in UFA, it is necessary to take into account that they could influence milk sensory quality and acceptance by consumers, mainly in two aspects (1) presence of off-flavors and (2) consumers’ milk overall acceptability.

It is noteworthy that, in this study, changes in milk FA profile of animals fed CPSC, with higher UFA, did not affect substantially the milk sensory assessment by the panel. Moreover, milk of cows fed CPSC had slightly improved flavor and overall acceptability than milk of CTR-fed cows. Regarding to milk sensory properties, flavor is one of the most important attributes for acceptability and, among the variables that determine milk flavor, fat is considered the most important one [36]. Focusing on the FA profile, the higher contents of stearic acid (C18:0) and PUFA and the reduction of palmitic acid (C16:0) in the milk from cows fed CPSC could promote the decrease of content of free FA, which are responsible for the reduction of sensory properties of milk [37]. In addition, the higher content in linoleic acid could enhance the proportion of hexanal, which is one of the volatiles most commonly present in raw and processed milks and is related with improved milk flavor [38].

However, it must be pointed out that these results are limited to pasteurized milk in a short-life storage time and more studies should be needed in order to know the stability in long-term storage systems.

## 5. Conclusions

Replacing hydrogenated palm fat with CPSC in the formulation of a concentrate for dairy cows, at the inclusion level tested in this study, results in milk with a more unsaturated FA profile, without observing a detrimental effect on digestibility, production performance, or milk acceptance.

CPSC is an attractive novel energetic feedstuff for livestock and a promising lipid supplement to change milk FA profile. In addition, CPSC can be obtained on the farm, making it a suitable local alternative which can contribute to the promotion of low-input production systems and with no competition with edible foods.

## Figures and Tables

**Figure 1 animals-09-00803-f001:**
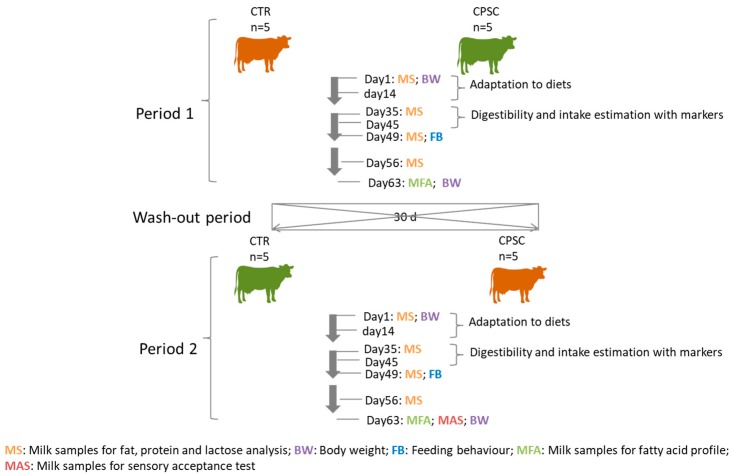
Scheme of the cross over experimental design and data frame of samplings.

**Table 1 animals-09-00803-t001:** Ingredients and chemical composition of basal diet and experimental concentrates (CTR, CPSC).

Item	Experimental Concentrates		
	CTR	CPSC	Basal Diet
Ingredients composition (g/kg DM)			
Corn	237	190	
Soybean meal	200	150	
Cold-pressed sunflower cake	0	230	
Palm kernel meal	150	0	
Destiled dry grains	149	10	
Barley	108	157	
Wheat	60	150	
Molasses	20	20	
Hydrogenated palm fat	20	0	
Alfalfa pellets	20	55	
Minerals and vitamins ^1^	36	38	
Maize silage			295
Grass silage			615
Barley straw			90
Chemical composition (g/kg DM)			
Dry matter	880	880	436
Starch	317	299	56
Crude protein	190	190	107
Neutral detergent fiber	225	195	411
Acid detergent fiber	97	98	336
Acid detergent Lignin	22	23	48
Fat	56	56	28

^1^ Contained (g/kg) calcium (270), magnesium (60), sodium (40), phosphorus (40) zinc (5.0), manganese (4.0), copper (1.5); (mg/kg), iodine (500), cobalt (50), selenium (15); (IU/g) retinyl acetate (500), cholecalciferol (100), DL-α-tocopheryl acetate (0.5); CTR: control; CPSC: cold-pressed sunflower cake.

**Table 2 animals-09-00803-t002:** Fatty acid composition of experimental concentrates (CTR, CPSC).

Item	CTR	CPSC
Key fatty acids (g/100 g of total fatty acids)		
C12:0	7.77	0.12
C13:0	0.03	0.02
C14:0	3.09	0.30
C150	0.04	0.04
C16:0	23.55	12.05
C17:0	0.08	0.08
C18:0	2.90	3.67
C20:0	0.33	0.34
C22:0	0.15	0.49
C23:0	0.06	0.08
C16:1 cis-9	0.12	0.16
C18:1 cis-9	25.39	20.56
C18:1 cis-11	1.13	1.59
C20:1 cis-11	0.22	0.22
C18:2 cis-9 cis-12	33.04	57.82
C18:3 cis-9 cis-12 cis-15	1.67	1.53

CTR: control; CPSC: cold-pressed sunflower cake.

**Table 3 animals-09-00803-t003:** Effect of feeding cold-pressed sunflower cake on intake, apparent digestibility of nutrients, and feeding behavior in lactating dairy cows (LSM; *n* = 10).

Item	Treatment		SED	*p*-Value		
	CTR	CPSC		Treatment	Breed	Period
DMI (kg/d)						
Roughage	15.4	15.2	0.76	0.867	0.260	0.491
Concentrate	4.7	4.7	0.22	0.958	0.352	0.512
Total	20.1	19.9	0.78	0.813	0.158	0.548
Total nitrogen	0.4	0.5	0.09	0.366	0.152	0.358
Starch	2.4	2.5	0.09	0.364	0.254	0.425
Neutral detergent fiber	9.6	9.7	0.44	0.947	0.138	0.421
Acid detergent fiber	5.5	5.6	0.25	0.921	0.301	0.540
Apparent digestibility (g/kg)						
Dry matter	763	745	20	0.381	0.964	0.136
Nitrogen	557	586	13	0.086	0.845	0.158
Starch	623	601	38	0.591	0.954	0.201
Neutral detergent fiber	544	528	22	0.503	0.899	0.150
Acid detergent fiber	460	430	30	0.342	0.950	0.155
Feeding behavior						
Intake, min/d	442	418	28.0	0.233	0.752	0.235
Rumination, min/d	372	418	46.8	0.177	0.835	0.321
Rumination, min/ kg DMI	19.9	20.2	1.43	0.579	0.852	0.299
Chewing, min/d	815	837	31.4	0.313	0.752	0.158
Chewing, min/kg DMI	43.4	40.4	2.74	0.137	0.780	0.221
Idling, min/d	624	602	31.4	0.313	0.899	0.450
Meal duration, min	42.5	35.2	6.41	0.129	0.952	0.423
Meals	9.3	9.7	1.72	0.690	0.901	0.520

DMr: roughage dry matter; DMc: concentrate dry matter; DMt: Total dry matter; CTR: control; CPSC: cold-pressed sunflower cake.

**Table 4 animals-09-00803-t004:** Effect of feeding cold-pressed sunflower cake on milk yield and composition of lactating dairy cows (LSM, *n* = 10).

Item	Treatments		SED	*p*-Value			
	CTR	CPSC		Treatment	Breed	Period	Week
Milk yield, kg/d	21.1	20.7	0.33	0.233	0.764	<0.001	<0.001
Milk/DMI	1.09	1.06	0.04	0.472	0.521	<0.001	<0.001
Milk fat, %	4.00	3.70	0.18	0.101	0.013	0.005	0.001
Yield, kg/d	0.86	0.78	0.03	0.013	0.025	0.004	0.001
3.5% FCM, kg/d	24.6	22.2	0.73	0.013	0.415	<0.001	<0.001
3.5% FCM/DMI	1.24	1.12	0.05	0.058	0.321	<0.001	<0.001
Milk crude protein, %	3.00	3.20	0.11	0.267	0.012	0.254	0.135
Yield, kg/d	0.65	0.66	0.04	0.816	0.045	0.125	0.258
N yield, g/d	102	103	5.7	0.816	0.247	<0.001	<0.001
As % of N intake	24.5	23.3	2.98	0.694	0.369	<0.001	<0.001
Milk lactose, %	4.8	4.9	0.05	0.342	0.350	0.488	0.352
Yield, kg/d	1.06	1.04	0.03	0.537	0.485	0.658	0.425

DMI = dry matter intake; FCM = fat corrected milk; CPSC: cold-pressed sunflower cake; CTR: control.

**Table 5 animals-09-00803-t005:** Effect of feeding cold-pressed sunflower cake on milk fatty acid profile of lactating dairy cows (LSM, *n* = 10).

Item	Treatments		SED	*p*-Value		
	CTR	CPSC		Treatment	Breed	Period
FA (g/100 g FA)						
C4:0	5.57	5.75	0.26	0.494	0.341	0.001
C6:0	2.22	2.29	0.08	0.452	0.876	0.081
C8:0	0.94	0.95	0.03	0.734	0.476	<0.001
C10:0	1.95	1.96	0.07	0.863	0.333	<0.001
C11:0	0.02	0.02	0.03	0.509	0.954	0.004
C12:0	2.62	2.37	0.09	0.023	0.299	<0.001
C13:0	0.06	0.07	0.01	0.145	0.283	0.005
C14:0 iso	0.12	0.14	0.02	0.241	0.396	0.002
C14:0	9.86	9.63	0.26	0.389	0.331	<0.001
C15:0 anteiso	0.51	0.51	0.01	0.956	0.456	<0.001
C15:0	0.95	1.01	0.04	0.147	0.269	<0.001
C16:0	30.7	29.1	0.64	0.035	0.628	<0.001
C17:0	0.51	0.52	0.02	0.520	0.038	<0.001
C18:0	11.5	12.2	0.28	0.025	0.708	<0.001
C18:1 cis-9	22.9	23.0	0.65	0.983	0.884	<0.001
C18:1 cis-11	0.29	0.29	0.03	0.953	0.983	<0.001
C18:1 cis-12	0.20	0.27	0.02	0.006	0.060	0.003
C18:1 trans-6	0.52	0.53	0.02	0.575	0.067	0.827
C18:1 trans-10	0.30	0.32	0.02	0.215	0.062	0.342
C18:1 trans-11	1.44	1.90	0.12	0.005	0.049	0.012
C18:1 trans-12	0.27	0.31	0.01	0.021	0.102	0.100
C18:2 trans-9 trans-12	0.03	0.04	0.01	0.324	0.034	<0.001
C18:2 cis-9 cis-12	1.71	1.93	0.05	0.004	0.139	<0.001
C18:2 cis-9 trans 11 CLA	0.65	0.88	0.06	0.004	0.067	0.097
C18:3 cis9 cis-12 cis-15	0.38	0.39	0.02	0.824	0.228	<0.001
C18:3n-6	0.03	0.03	0.01	0.170	0.336	0.391
C20:0	0.19	0.21	0.02	0.353	0.099	0.006
C20:1n-9 cis-11	0.04	0.04	0.01	0.889	0.754	0.647
C20:2n-6	0.02	0.02	0.01	0.968	0.277	0.206
C20:3n-6	0.08	0.09	0.01	0.486	0.712	0.048
C20:3n-3	0.11	0.12	0.01	0.452	0.229	0.007
C20:4n-6	0.02	0.02	0.01	0.632	0.952	0.005
C21:0	0.04	0.05	0.01	0.117	0.135	0.007
C22:2n-6	0.04	0.04	0.01	0.641	0.636	<0.001
C22:5n-3	0.07	0.07	0.01	0.525	0.268	0.009
C22:6n-3	0.01	0.01	0.01	0.901	0.067	0.059
C23:0	0.04	0.05	0.01	0.104	0.307	0.028
C24:0	0.07	0.09	0.01	0.173	0.229	0.003
C24:1n-9	0.01	0.01	0.01	0.019	0.238	0.466
∑BCFA	0.86	0.90	0.05	0.423	0.323	<0.001
∑SFA	68.73	67.82	0.75	0.238	0.645	<0.001
∑ cis MUFA	26.0	25.90	0.66	0.877	0.833	<0.001
∑ trans MUFA	2.52	3.05	0.13	0.003	0.010	0.016
∑MUFA	28.5	29.00	0.71	0.570	0.874	<0.001
∑PUFA	3.32	3.81	0.14	0.006	0.046	0.381
∑CLA	0.81	1.06	0.07	0.007	0.082	0.215
C18:1 trans-10/trans-11	0.22	0.18	0.02	0.173	0.053	0.032
n-6:n-3	3.67	3.86	0.16	0.279	0.733	<0.001
PUFA:SFA	0.05	0.06	0.01	0.006	0.091	0.039

FA: fatty acids; BCFA: branched-chain fatty acids; SFA: saturated fatty acid; MUFA: monounsaturated fatty acids; PUFA: polyunsaturated fatty acids; CLA: conjugated linoleic acid; CTR: control; CPSC: cold-pressed sunflower cake.

**Table 6 animals-09-00803-t006:** Effect of feeding cold-pressed sunflower cake on milk sensorial quality of lactating dairy cows.

Item	Treatment		SED	*p*-Value	
	CTR	CPSC		Treatment	Breed
Overall acceptability	5.9	6.5	0.19	0.003	0.112
Appearance	6.7	6.7	0.17	1.000	0.921
Odor	5.9	5.9	0.18	0.702	0.444
Texture	6.6	6.4	0.55	0.629	0.149
Flavor	6.0	6.6	0.22	0.008	0.049

CTR: control; CPSC: cold-pressed sunflower cake.

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
