# Peer review of "Use of Cold-Pressed Sunflower Cake in the Concentrate as a Low-Input Local Strategy to Modify the Milk Fatty Acid Profile of Dairy Cows"

_animals, 2019, doi:10.3390/ani9100803_

Round 1

Reviewer 1 Report

General Comments

The manuscript “Use of cold-pressed sunflower cake in the concentrate as a low-input local strategy to modify the milk fatty acid profile of dairy cows” describes the effect of cold-pressed sunflower cake supplementation on mid-lactating dairy cows on milk fatty acids profile, performance and milk sensory properties. I really enjoyed reading this manuscript. The quality of English is overall excellent, and the authors well described the experiment and results.

However, some minor comments are reported below in order to improve the manuscript with further data and in particular reporting a figure to show the experimental design, helping readers to quickly understand the study.

Specific comments:

Table 1. I would suggest reporting CTR and CPSC diet including also the forages. For me and I think for readers as well would be easier to quickly understand the diets, ingredients, and chemical composition. Doing this, you will have the chemical composition of the whole diet, instead to have separately the concentrate and forage-based chemical composition.

L93-99: I suggest the authors to report a figure with the time frame of the experimental design, indicating the adaptation period, the experimental period and observations for intake, milk production, and samples.

L95: Please delete “Place here”

L136: report the reference name.

L156-157: Did the authors composite milk samples relatively to milk production?

L158: For FA profile, did authors proceed on individual milk samples or composited milk samples?

L201: Please, could you spell DMt and DMr? They are reported in the table, but I suggest spelling them here since it is the first time.

Table 3: Authors reported to have concentrate intake data. I suggest adding them here in the table. Estimation of Intake data sounds weird to me. The total DMI is kind in accordance with the average DMI of cows producing no more than 30 kg/d. However, the intake of starch seems really low. From the other side I understand that most of starch comes from maize silage and considering the fact that forage intake was estimated by fecal-output indigestibility technique, this could really bias the overall intake. Despite this limitation, again I suggest reporting the data on concentrate intake.

Table 3, 4, 5, 6, and 7. Could authors add the P value of period, week, and breed? Also, the P should be Italicized throughout the text.

Author Response

Reviewer 1

General Comments

The manuscript “Use of cold-pressed sunflower cake in the concentrate as a low-input local strategy to modify the milk fatty acid profile of dairy cows” describes the effect of cold-pressed sunflower cake supplementation on mid-lactating dairy cows on milk fatty acids profile, performance and milk sensory properties. I really enjoyed reading this manuscript. The quality of English is overall excellent, and the authors well described the experiment and results.

However, some minor comments are reported below in order to improve the manuscript with further data and in particular reporting a figure to show the experimental design, helping readers to quickly understand the study.

We want to thanks the time spent in revising our manuscript; your suggestions have been very helpful to improve the document. 

Specific comments:

Table 1. I would suggest reporting CTR and CPSC diet including also the forages. For me and I think for readers as well would be easier to quickly understand the diets, ingredients, and chemical composition. Doing this, you will have the chemical composition of the whole diet, instead to have separately the concentrate and forage-based chemical composition

We understand the point of view of the reviewer, but the ration consisted of a forage mix offered ad libitum and a concentrate individually offered sepatedly and measured, so a priori we cannot know what is the forage:concentrate ratio. If we have to report the chemical composition of the whole diet as requested by reviewer, we have to assume a forage-to-concentrate ratio a priori. Therefore we think that is clearer for the readers to report the chemical composition as appeared in the paper.

L93-99: I suggest the authors to report a figure with the time frame of the experimental design, indicating the adaptation period, the experimental period and observations for intake, milk production, and samples.

A Figure with the time frame of the experimental design was included in the manuscript in L 110

L95: Please delete “Place here”

Place here” was removed

L136: report the reference name.

Reference name was included in L163

L156-157: Did the authors composite milk samples relatively to milk production?

Yes milk samples were composited relatively to milk production; It was included in L189-190

L158: For FA profile, did authors proceed on individual milk samples or composited milk samples?

We composited the samples relatively to milk production, but on an individual basis (one sample per cow)

L201: Please, could you spell DMt and DMr? They are reported in the table, but I suggest spelling them here since it is the first time.

We have spelt this concepts in a clearer way in the text (L242-243) and table 3

Table 3: Authors reported to have concentrate intake data. I suggest adding them here in the table. Estimation of Intake data sounds weird to me. The total DMI is kind in accordance with the average DMI of cows producing no more than 30 kg/d. However, the intake of starch seems really low. From the other side I understand that most of starch comes from maize silage and considering the fact that forage intake was estimated by fecal-output indigestibility technique, this could really bias the overall intake. Despite this limitation, again I suggest reporting the data on concentrate intake.

Data on concentrate intake was included in Table 3

Table 3, 4, 5, 6, and 7. Could authors add the P value of period, week, and breed? Also, the P should be Italicized throughout the text.

P values of period, breed and week have been included in Table 4.

In Table 3 and 5 (in the previous version of the manuscript Table 5 and 6) P values of period and breed were included, but.no week effect was included because, as was described in statistical analysis section, week effect was not included to analyse these data.

In Table 6 (Table 7 in the previous version of the manuscript) only breed effect was included, since samples were taken only on Period 2 as described in Material and Methods section (L234-236)

P has been Italicized throughout the text

Reviewer 2 Report

The manuscript animals-605308 was carried out to investigate if CPSC could be used in the formulation of a concentrate for dairy cows, totally replacing hydrogenated palm fat, with the advantage of improving milk FA profile. It is technically sound. Some details are needed probably because this paper is in the interface between dairy production and food science and sometimes authors tend to explain with more detail one of these angles. The main flaws are when the authors set up the scene for objectives and hypothesis.

Line 42 add reference and make sure you mention which animal species milk you are talking about; it could be in ruminants or in specific animals…

Line 43 add reference

Line 51 add a more detailed description of cold-pressing processing. Also, please mention other similar technologies i.e., spry drying, etc. A non-expert reader would like to know if there are other alternatives for processing oilseed by products or oil industry by-products.

Lines 69-72 please add an objective for your research. A hypothesis is very well welcome but it will be clearer if an objective statement is written.

Line 85 add a description of the rationale behind the formulation of the control or basal TMR. Was it formulated from the NRC guidelines? What were the theoretical assumptions? Excepted milk production? Days in milk?

Missing information that needs to be added somewhere in section 2.1

Housing conditions Did you add the hydrogenated fat as top-dressed or it was added in the feeding wagon? If it was the latest, how did you ensure that the animals ate the exact amount of treatment? What was in % of DM the inclusion of supplemental fat added to the control diet? And how did you came up with that number? Due to previous experiments? Please mention the rationale behind that.

Table 1 please add the dry matter contents of concentrates and control diet. Please describe DDG, NDF, ADF, ADL no need for acronyms as you have enough space

Line 153 please mention if it was FAME 37???

Table 3 what is N? if it is nitrogen then explain which form of N compound i.e., total N, NH3-N, etc. Also, no need for acronyms as you have enough space

Line 177 please add as Supplemental material the description of the measured attributes

178 why did you analyze behavior?  You need to clear that out in the introduction and the methods for that need to be very well explain in section 2.3.3 with references or previous own-research.

Tables 5 and 6 need to be merged, splitting saturated from unsaturated is confusing and as reader, you miss the thread…. Please put all data together. Perhaps the main totals and ratios can be included in  different table.

Lines 268-272 yes, and that is why you need to explain the rationale behind the inclusion levels from your treatments.

Lines 355-356 re write as it is difficult to understand the massage otherwise delete. Also include some thoughts about the implications for your study for the dairy industry.

In the discussion section:

You mention in the title the word low-input local strategy but this is not well described in the discussion of your data. Please elaborate on that, as this is a pillar from your manuscript.

You have a very nice experiment that looked into the animal, the product and consumer perceptions. The problems is that you need to discuss all of those angles. In the present version, the messages are still too weak for the reader and I think that you have a very nice data set and you have spent money and efforts for this. Then, I will recommend that you go back and work on the key aspects from your experiments and make them crystal-clear for the readers.

For example, in your results there are very few variables with a significant change? Was that on purpose? – you need to explain that…

You have digestibility and milk FA, you need to link that to the lack of effects from your treatments and then link it to the perceived sensory characteristics of the final food matrix.

I really like your paper and the lack of significant effects is fine, but you really need to back it up in your discussions, otherwise, why will you need to buy this cold-pressing lipid supplement if a common TMR does the same??? Do you know what I mean?

Author Response

Reviewer 2

The manuscript animals-605308 was carried out to investigate if CPSC could be used in the formulation of a concentrate for dairy cows, totally replacing hydrogenated palm fat, with the advantage of improving milk FA profile. It is technically sound. Some details are needed probably because this paper is in the interface between dairy production and food science and sometimes authors tend to explain with more detail one of these angles. The main flaws are when the authors set up the scene for objectives and hypothesis.

We really appreciate reviewer´s comments; they have been be very useful to improve the paper

Line 42 add reference and make sure you mention which animal species milk you are talking about; it could be in ruminants or in specific animals…

We have included animal species (ruminants in this case) in L, and a reference

Line 43 add reference

We have added a reference in L45

Line 51 add a more detailed description of cold-pressing processing. Also, please mention other similar technologies i.e., spry drying, etc. A non-expert reader would like to know if there are other alternatives for processing oilseed by products or oil industry by-products.

A more detailed description of the cold-pressing processing has been included in L61-62

The other alternatives to process oilseeds is the conventional one produced after extracting the oil from seed using hexane combined with high temperatures, giving a cake with less oil and relatively low dietary energy value and a higher protein level, which can be used as a protein source. We have done a mention to this process in L64

Lines 69-72 please add an objective for your research. A hypothesis is very well welcome but it will be clearer if an objective statement is written.

An objective has been included in L81

Line 85 add a description of the rationale behind the formulation of the control or basal TMR. Was it formulated from the NRC guidelines? What were the theoretical assumptions? Excepted milk production? Days in milk?

Animals were not fed with a TMR, they were group fed a basal forage mixed ration ad libitum and individually fed a fixed quantity of concentrate (CTR or CPSC). Rations were formulated to cover lactation needs taking into account milk production and days in milk according to INRA tables. Explanations about rationale behind the formulation have been included in L99-101

Missing information that needs to be added somewhere in section 2.1

Housing conditions Did you add the hydrogenated fat as top-dressed or it was added in the feeding wagon? If it was the latest, how did you ensure that the animals ate the exact amount of treatment? What was in % of DM the inclusion of supplemental fat added to the control diet? And how did you came up with that number? Due to previous experiments? Please mention the rationale behind that.

The hydrogenated palm fat was included in the formulation of the commercial concentrate. The “treatment” was formulated into de concentrate; concentrate DM intake was recorded and was not different between treatments, so we can infer that animals ate the exact amount of treatment.

The percentage of fat of the two diets was the same, the CTR concentrate has hydrogenated palm fat and the CPSC has sunflower oil contained in it as fat source. The percentage of fat of both diets was the same: 5,6% of the concentrate (CTR and CPSC) (Table1) and approximately 4.5 % of total diet (taking into account forage intake level estimated with markers). We formulated the concentrate taking into account the forage to concentrate ratio normally observed in the experimental farm of Fraisoro and taking into account not to include a fat content in total diet more than 5% that can be related with depressed milk fat content or milk production, especially with fat sources rich in polyunsaturated fatty acids.

Table 1 please add the dry matter contents of concentrates and control diet. Please describe DDG, NDF, ADF, ADL no need for acronyms as you have enough space

We have added DM of concentrate and forages in Table 1

We have deleted acronyms and write the whole names in Table 1

Line 153 please mention if it was FAME 37???

We used Supelco37 FAME mixed (Sigma Aldrich) and the Nu-Chek Prep mix were used to identify the FAME not contained in the previous FAME mix.

Table 3 what is N? if it is nitrogen then explain which form of N compound i.e., total N, NH3-N, etc. Also, no need for acronyms as you have enough space

Table 3 has been changed as requested

Line 177 please add as Supplemental material the description of the measured attributes

We made the milk sensory acceptance test with a non trained panel. These consumers were asked to describe their impression in terms of intensity with a 9 points scale. They were not a trained panel and were not asked to identify specific characteristics such as Bitter, melty…

Therefore we have not included the supplemental material suggested

178 why did you analyze behavior?  You need to clear that out in the introduction and the methods for that need to be very well explain in section 2.3.3 with references or previous own-research.

We analysed feeding behaviour because any feeding strategy that could modified digestibility and intake, like adding polyunsaturated fats to the diet, could also modified feeding behaviour. The changes occurred in time spent eating or ruminating for example could help us to understand the changes observed in digestibility or intake. In this study no differences were observed (that is fine) both in digestibility or feeding behaviour, but we think it is important to assess that no disturbances occurred in this traits, and that is why we included it in the study.

We included a brief explanation in discussion section in L388-391

Tables 5 and 6 need to be merged, splitting saturated from unsaturated is confusing and as reader, you miss the thread…. Please put all data together. Perhaps the main totals and ratios can be included in  different table.

Tables 5 and 6 have been merged as requested

Lines 268-272 yes, and that is why you need to explain the rationale behind the inclusion levels from your treatments.

This issue has been replied in a previous point.

Lines 355-356 re write as it is difficult to understand the massage otherwise delete. Also include some thoughts about the implications for your study for the dairy industry.

We have deleted this sentence

An implication for the dairy industry has been included in L434-438

In the discussion section:

You mention in the title the word low-input local strategy but this is not well described in the discussion of your data. Please elaborate on that, as this is a pillar from your manuscript.

We include new information in the discussion section (L314-326)

You have a very nice experiment that looked into the animal, the product and consumer perceptions. The problems is that you need to discuss all of those angles. In the present version, the messages are still too weak for the reader and I think that you have a very nice data set and you have spent money and efforts for this. Then, I will recommend that you go back and work on the key aspects from your experiments and make them crystal-clear for the readers.

For example, in your results there are very few variables with a significant change? Was that on purpose? – you need to explain that…

You have digestibility and milk FA, you need to link that to the lack of effects from your treatments and then link it to the perceived sensory characteristics of the final food matrix.

I really like your paper and the lack of significant effects is fine, but you really need to back it up in your discussions, otherwise, why will you need to buy this cold-pressing lipid supplement if a common TMR does the same??? Do you know what I mean?

The key aspects of the experiment are that CPSC can be used as an ingredient in the concentrate for dairy cows without affecting negatively digestion, intake production or product sensory qualities, and that the use of this ingredient can modify milk FA profile to obtain healthier milk. We think these aspects are reflected in the manuscript.

Other key aspect is the contribution of this feeding strategy to low input systems in dairy farms and the no competition of this ingredient with human edible foods. This aspect was not clear explained in the previous version and we added some related information in the discussion section and in the conclusions.

We expect that the included changes could contribute to answer the reviewer´s questions and doubts.

Round 2

Reviewer 2 Report

Authors have done corrections accordingly buy they just need to add properly, the information below in the materials and methods section. After this is done then the manuscript is ready to be considered for publication.

The hydrogenated palm fat was included in the formulation of the commercial concentrate. The “treatment” was formulated into de concentrate; concentrate DM intake was recorded and was not different between treatments, so we can infer that animals ate the exact amount of treatment.
The percentage of fat of the two diets was the same, the CTR concentrate has hydrogenated palm fat and the CPSC has sunflower oil contained in it as fat source. The percentage of fat of both diets was the same: 5,6% of the concentrate (CTR and CPSC) (Table1) and approximately 4.5 % of total diet (taking into account forage intake level estimated with markers). We formulated the concentrate taking into account the forage to concentrate ratio normally observed in the experimental farm of Fraisoro and taking into account not to include a fat content in total diet more than 5% that can be related with depressed milk fat content or milk production, especially with fat sources rich in polyunsaturated fatty acids.

Author Response

We have addressed all the points in the new document in Line 107-113. Thanks a lot for improving the manuscript